# Detecting behavioural changes in human movement to inform the spatial scale of interventions against COVID-19

**Hamish Gibbs**[1]*, **Emily Nightingale**[2], **Yang Liu**[1], **James Cheshire**[3], **Leon Danon**[4,5,6], **Liam Smeeth**[7], **Carl A. B. Pearson**[1], **Chris Grundy**[1], **LSHTM CMMID COVID-19 working group**[1¶], **Adam J. Kucharski**[1], **Rosalind M. Eggo**[1]*

1 Department of Infectious Disease Epidemiology, London School of Hygiene & Tropical Medicine, London, United Kingdom, 2 Department of Global Health and Development, London School of Hygiene & Tropical Medicine, London, United Kingdom, 3 Department of Geography, University College London, London, United Kingdom, 4 Department of Engineering Mathematics, University of Bristol, Bristol, United Kingdom, 5 The Alan Turing Institute, British Library, London, United Kingdom, 6 Population Health Sciences, University of Bristol, Bristol, United Kingdom, 7 Faculty of Epidemiology and Population Health, London School of Hygiene & Tropical Medicine, London, United Kingdom

¶ Membership of LSHTM CMMID COVID-19 working group is provided in the acknowledgments.
* hamish.gibbs@lshtm.ac.uk (HG); r.eggo@lshtm.ac.uk (RME)

**Data Availability Statement:** Mobility Data used in this study are proprietary data owned by Facebook. The data used in this study is are not publicly available but are available to researchers by

## Abstract

On March 23 2020, the UK enacted an intensive, nationwide lockdown to mitigate transmission of COVID-19. As restrictions began to ease, more localized interventions were used to target resurgences in transmission. Understanding the spatial scale of networks of human interaction, and how these networks change over time, is critical to targeting interventions at the most at-risk areas without unnecessarily restricting areas at low risk of resurgence. We use detailed human mobility data aggregated from Facebook users to determine how the spatially-explicit network of movements changed before and during the lockdown period, in response to the easing of restrictions, and to the introduction of locally-targeted interventions. We also apply community detection techniques to the weighted, directed network of movements to identify geographically-explicit movement communities and measure the evolution of these community structures through time. We found that the mobility network became more sparse and the number of mobility communities decreased under the national lockdown, a change that disproportionately affected long distance connections central to the mobility network. We also found that the community structure of areas in which locally-targeted interventions were implemented following epidemic resurgence did not show reorganization of community structure but did show small decreases in indicators of travel outside of local areas. We propose that communities detected using Facebook or other mobility data be used to assess the impact of spatially-targeted restrictions and may inform policy-makers about the spatial extent of human movement patterns in the UK. These data are available in near real-time, allowing quantification of changes in the distribution of the population across the UK, as well as changes in travel patterns to inform our understanding of the impact of geographically-targeted interventions.

application to the Facebook Data for Good Partner Program (https://dataforgood.fb.com/, diseaseprevmaps@fb.com). Census data from national statistics agencies is publicly available. Dataset information is available in the references. Code used to conduct this analysis is available from https://github.com/cmmid/uk_movement_communities_covid.

**Funding:** The following funding sources are acknowledged as providing funding for the named authors. This research was partly funded by the Bill & Melinda Gates Foundation (https://www.gatesfoundation.org/) (INV-003174: YL; NTD Modelling Consortium OPP1184344: CABP; OPP1183986: EN). DFID/Wellcome Trust (https://www.gov.uk/government/organisations/foreign-commonwealth-development-office, https://wellcome.org/) (Epidemic Preparedness Coronavirus research programme 221303/Z/20/Z: CABP). This project has received funding from the European Union's Horizon 2020 research and innovation programme - project EpiPose (https://cordis.europa.eu/project/id/101003688) (101003688: YL). HDR UK (https://www.hdruk.ac.uk/) (MR/S003975/1: RME). UK DHSC/UK Aid/NIHR (https://www.gov.uk/government/organisations/department-of-health-and-social-care, https://www.ukaiddirect.org/, https://www.nihr.ac.uk/) (PR-OD-1017-20001: HPG). UK MRC (https://mrc.ukri.org/) (MC_PC_19065: RME, YL). Wellcome Trust (https://wellcome.org/) (206250/Z/17/Z: AJK), UK MRC (https://mrc.ukri.org/) (MC_PC_19067, MR/V038613/1 LD), UK EPSRC (https://epsrc.ukri.org/) (EP/V051555/1: LD), The Alan Turing Institute under the EPSRC (https://www.turing.ac.uk/) (grant EP/N510129/1: LD). This research was partly funded by the National Institute for Health Research (NIHR) using UK Aid from the UK Government to support global health research. The views expressed in this publication are those of the author(s) and not necessarily those of the NIHR or the UK Department of Health and Social Care (16/137/109: YL; NIHR200908: RME). The funders had no role in study design, data collection and analysis, decision to publish, or preparation of the manuscript. The following funding sources are acknowledged as providing funding for the working group authors. Alan Turing Institute (AE). BBSRC LIDO (BB/M009513/1: DS). This research was partly funded by the Bill & Melinda Gates Foundation (INV-001754: MQ; INV-003174: KP, MJ; NTD Modelling Consortium OPP1184344: GFM; OPP1180644: SRP; OPP1191821: KO'R, MA). BMGF (OPP1157270: KA). DFID/Wellcome Trust (Epidemic Preparedness Coronavirus research programme 221303/Z/20/Z: KvZ). DTRA (HDTRA1-18-1-0051: JWR). Elrha R2HC/UK DFID/

## Author summary

Large-scale intensive interventions in response to the COVID-19 pandemic have been implemented globally, significantly affecting human movement patterns. Mobility data provide an empirical measurement of UK travel within a spatially-explicit network, but it is not clear how the structure of that network changed in response to national or locally-targeted interventions. We use daily mobility data aggregated from Facebook users to quantify changes in the travel network in the UK during the national lockdown, and in response to local interventions. We identified changes in human behaviour in response to interventions and identified the community structure inherent in these networks. This approach can help quantify the extent of strongly connected communities of interaction and their relationship to the extent of spatially-explicit interventions. We show that spatial mobility data available in near real-time can give information on connectivity that can be used to understand the impact of geographically-targeted interventions and in the future, to inform these intervention strategies.

## Introduction

Fine-scale geographic monitoring of large populations can potentially increase the accuracy and responsiveness of epidemiological modelling, outbreak response, and intervention planning in response to public health emergencies like the COVID-19 pandemic [1–6]. Population and mobility datasets collected from the movement of individuals' mobile phones provide empirical, near-real time metrics of population movement between different geographic regions [6–8]. The COVID-19 pandemic response has coincided with the availability of new data sources for measuring human movement, aggregated from mobile devices by network providers and popular applications including Google Maps, Apple Maps, Citymapper, and Facebook [7,9,10].

The spread of COVID-19 through travel networks has been demonstrated in China, where connectivity to Wuhan was shown to predict the timing of arrival of COVID-19 cases in each region [11,12]. The volume of mobility has also been shown to correlate with transmission of COVID-19, where movement can be used as a proxy for the measurement of the degree of social distancing [13]. Travel and movement behavior during epidemics may also change in response to imposed interventions, perceived risk, and due to seasonal activities such as vacations [11,12]. During the COVID-19 pandemic, mobility data has been used to assess adherence to movement restrictions [13,14], the impact of movement restrictions on the transmission dynamics of COVID-19 [15–17], and demonstrate differential adherence to movement restrictions among socioeconomic groups [18–21].

In this analysis, we use movement and population data provided by Facebook from March 10 to November 1 2020, which records approximately 15 million daily locations of 4.8 million users [7]. We also used population, age, ethnicity, and Index of Multiple Deprivation (IMD) data from UK national statistics agencies to understand the population of users recorded in the movement and population data [22–25]. We identify changes in travel behavior in response to initially stringent movement restrictions (March to May 2020) and subsequent easing of restrictions, paired with a policy of spatially-targeted interventions in response to local resurgences (May to October 2020). Using network analytic methods to understand the structure of interconnected communities in the movement network, we trace the evolution of

Wellcome Trust/This research was partly funded by the National Institute for Health Research (NIHR) using UK aid from the UK Government to support global health research. The views expressed in this publication are those of the author(s) and not necessarily those of the NIHR or the UK Department of Health and Social Care (KvZ). ERC Starting Grant (#757699: JCE, MQ, RMGJH). This project has received funding from the European Union's Horizon 2020 research and innovation programme - project EpiPose (101003688: KP, MJ, PK, RCB, WJE). This research was partly funded by the Global Challenges Research Fund (GCRF) project 'RECAP' managed through RCUK and ESRC (ES/P010873/1: AG, CIJ, TJ). MRC (MR/N013638/1: NRW). Nakajima Foundation (AE). NIHR (16/136/46: BJQ; 16/137/109: BJQ, CD, FYS, MJ; Health Protection Research Unit for Immunisation NIHR200929: NGD; Health Protection Research Unit for Modelling Methodology HPRU-2012-10096: TJ; NIHR200929: FGS, MJ; PR-OD-1017-20002: AR, WJE). Royal Society (Dorothy Hodgkin Fellowship: RL; RP\EA\180004: PK). UK MRC (LID DTP MR/N013638/1: GRGL, QJL; MC_PC_19065: AG, NGD, SC, TJ, WJE; MR/P014658/1: GMK). Authors of this research receive funding from UK Public Health Rapid Support Team funded by the United Kingdom Department of Health and Social Care (TJ). Wellcome Trust (206250/Z/17/Z: TWR; 206471/Z/17/Z: OJB; 208812/Z/17/Z: SC, S Flasche; 210758/Z/18/Z: JDM, JH, KS, NIB, SA, SFunk, SRM). No funding (AKD, AMF, AS, CJVA, DCT, JW, KEA, SH, YJ, YWDC).

**Competing interests:** The authors have declared that no competing interests exist.

geographic communities through time, comparing them to intervention measures implemented in response to local resurgences.

## Materials and methods

### Ethics statement

This research was approved by the LSHTM Observational Research Ethics Committee (ref 16834–1).

### Facebook data

Data provided by the Facebook Data for Good partner program [7] uses aggregated and anonymized user data to record user locations in grid cells (S1 Fig). This data is generated from the population of Facebook users with location services actively enabled and is released approximately 48 hours after data collection.

We used movement data, describing users' modal location in map cells in sequential 8 hour periods. This means that, in each period, a user is assigned to the cell in which they record the largest number of locations. For this individual, the beginning and end points of a network edge are defined as their location in sequential periods. This individual's movement is then aggregated with others to form a weighted Origin-Destination matrix of movement between grid cells. Edges with fewer than 10 travellers were removed by Facebook prior to data sharing to preserve privacy (S2 Fig). Any cell that did not record any between-cell travels with greater than 10 travellers in a given time window was omitted from the dataset, regardless of whether that cell recorded an internal number of users greater than 10. In our network analysis, we constructed a weighted, directed network where nodes were cells, and edge weights were the number of users observed travelling between cells.

We use movement outside of local areas as a measurement of movement from one cell to any other cell in the network. The movement data also records connections from one cell to the same cell in sequential periods. This type of movement may indicate movement within a local area (less than the area of a given cell), or completely stationary individuals. Because of the spatial resolution of the data, it is not possible to quantify the volume of within cell movement that corresponds to each of these behaviours within the same cell. We therefore use the volume of movement outside of the cell as a measurement of the overall movement behaviour, making the assumption that travellers leaving a cell will primarily return to the same cell in sequential periods.

### Bing maps tile system

Movement data is referenced to the Bing Maps Tile System, a standard geospatial reference used primarily for serving web maps [26]. The system is divided into 23 zoom levels ranging from global level 1, (map scale: 1:295,829,355.45) to detailed level 23, (map scale: 1:70.53). Each Bing Map cell is identified by a "quadkey", or unique identifier of the zoom level and pixel coordinates of an individual cell. In this analysis, all mobility and census datasets were referenced to Bing Maps cells. The movement dataset was referenced to cells at zoom level 12 (approximately 4.8 to 6.2 $km^2$ in the UK—measured at 60.77˚ and 50.59˚ respectively). The ground resolution of cells varies with latitude, with cells at higher latitudes covering a smaller ground area than those at lower latitudes. This distortion is caused by the distortion inherent in the Web Mercator projection (EPSG:3857) used by the Bing Maps Tile System.

## Demographic information

We compared the age, population, ethnicity, and IMD of each cell to the population of users recorded in the movement data to identify relationships between the percentage of Facebook users and these demographic factors [27–32]. We extracted these variables from national statistics agencies (Office for National Statistics, Northern Ireland Statistics and Research Agency, Scottish Government, and Welsh Government) and aggregated them to grid cells. Census variables were referenced to different statistical units by country. In Northern Ireland, census variables were referenced to Super Output areas (SOAs), in England and Wales, Lower Super Output areas (LSOAs), and in Scotland, Data Zones (DZs). Detailed population data was also collected from national statistics agencies, providing a measure of population for Small Areas (Northern Ireland), Output Areas (OA; England and Wales), and Data Zones (Scotland) [33].

Census variables referenced to different national statistical areas were aggregated to align with mobility datasets at zoom level 12. First, we combined the 2011 population weighted centroids of each OA (or equivalent) from the UK Census with 2020 mid-year population estimates in each UK country. We then assigned each OA centroid to the cell it falls within. We then joined 2011-derived census variables (Age, Ethnicity and IMD) to the OA centroids and computed an average of each census variable for each cell, weighted by the OA population estimates. For IMD data (recorded as ranks) we ranked the weighted average values to create a rank of cells by their population weighted IMD. OAs are much more granular than cells and therefore nested within them in the majority of cases, minimising the risk of the cells detrimentally intersecting OAs during the spatial assignment of demographic variables.

To assess the correlation between census variables and the proportion of Facebook users in each cell, we computed the Pearson correlation coefficient and two-sided p-values between the proportion of Facebook users in a cell and each census variable.

## Temporal aggregation

Both Facebook movement and population data are recorded in 8 hour intervals. These data display strong and consistent intraday and intraweek patterns. To isolate changes in daily mobility, data collected in 8 hour periods were aggregated to daily periods by taking the sum of the observed number of travellers along an edge for all periods within a day.

## Baseline population estimates

To obtain an accurate measurement of the number of users in the Facebook movement and population datasets relative to census population estimates, we used baseline measurements of travel from one cell to the same cell in sequential time periods computed during the 45 days prior to the creation of the data collection, from January 29th to March 9th 2020. This baseline population recorded the median number of users in a cell for each daily time window in the reference period.

## Edge betweenness centrality

We computed the edge betweenness centrality of the movement network, a measure of which edges are most responsible for maintaining connection between different parts of the network. Betweenness centrality was calculated for the weighted, directed network. Comparing this measure of centrality provides information about changes in network topology, the volume of travel along the network, and which connections might be preferred targets for interventions.

## Community detection

Community detection methods are algorithms for identifying groups of meaningfully connected vertices in a network. Many methods exist, with various tradeoffs on computational performance, resolution, or other characteristics [34–37]. Different community detection methods produce different results because of differences in the network characteristics that they use to define communities. To understand the robustness of the communities detected in this study, we employed two different algorithms, InfoMap and Leiden. InfoMap is a community detection algorithm based on the Map equation which records the movements of a random walker along the network. The algorithm tests network partitions, attempting to identify the partition that minimizes the description length required to describe the walker's path [38]. In other words, the algorithm compresses the description of movements in parts of the network where the walker spends a large amount of time, thereby forming communities from these strongly connected sections of the network. The Leiden algorithm maximizes the modularity of different node partitions. To do this, the algorithm assigns network nodes to a partition, assesses the modularity of the partitioned groups, aggregates the nodes in a given partition, and repeats this process until there is no further improvement in the modularity of the partitions. This process results in a partition in which communities possess stronger connections to members of their own community than to other network nodes [39,40]. There is a penalty for both algorithms if they identify a number of communities larger than the minimum number required to maximize their respective objective functions (InfoMap: description length, Leiden: modularity).

We compared the effect of the different community detection algorithms, and found that they aligned hierarchically, where the Leiden algorithm identified geographically larger communities. If the communities detected by one method are largely a superset of the communities detected by another, with shared boundaries between the defined communities, this likely represents a differing hierarchical structure, compared to a different interpretation of community structure. We assessed the agreement between community detection methods to understand the stability of detected communities by comparing the proportion of nodes in each community detected using InfoMap with all communities determined using Leiden, and vice versa (S9 Fig). This comparison allows for the computation of the proportion of shared nodes between both algorithms. The maximum and mean overlap of communities in each algorithm helped to identify the agreement between each method of community detection. In general we found that Leiden detected larger communities, for which the InfoMap communities were (for the most part) sub-communities.

## Community label inheritance

The community detection methods used in this study identified communities each day. To track the evolution of communities over the study period, we employed a heuristic approach, assigning the label of a given community identified in a certain time step to that community with the highest number of shared nodes in the following time step [41]. When multiple communities in a certain time step "claim" the same community in the following timestep, the community with the closest size to the community in the following timestep "wins" the right to pass its own label to the following timestep.

## COVID-19 data

We used confirmed COVID-19 cases from the UK Pillar 1 and Pillar 2 testing schemes available for England [42]. Pillar 1 is predominantly hospital-based tests including patients and health care workers. Pillar 2 is symptomatic community testing on demand, and represents the

bulk of the testing in the UK. Data on the number of confirmed SARS-CoV-2 positive tests by specimen date were available at the Lower Tier Local Authority (LTLA) level [43].

To compare confirmed COVID-19 cases to movement indicators, we measured the total proportion of travellers leaving a grid cell in monthly periods for all cells in England. Cells were then assigned to LTLAs by their maximum areal overlap.

## Results

### Movement patterns observed

To understand the representativeness of Facebook data to the general population, we explored the size of the population of Facebook users included in the movement dataset, and compared this population to 2019 UK census population estimates. Facebook recorded an average of 4.5 million users per 8 hour period, ranging from 5.8 million on March 29th between 4pm and midnight, to 3.7 million on August 9th between midnight and 8am (Fig 1A).

The percentage of Facebook users per grid cell was comparable in the four nations of the UK (Fig 1B). The population of Facebook users was highly correlated (R = 0.97, p < 2.2e-16) with census population estimates for all cells (Fig 1C) although there is variance in the proportion of Facebook users to census population across cells, t (Fig 1D). We compared the proportion of Facebook users to identify whether the distribution of Facebook users was biased in relation to a specific demographic variable, finding no strong associations between the percentage of Facebook users and the average age, percent minority ethnic, population density, or IMD of each cell (S3 and S5 Figs).

Using COVID-19 case data at LTLA level in England, we identified a consistent association between the proportion of users travelling outside of grid cells and the number of cases in LTLAs per month during the study period (Figs 2 and S6). While the strength of this association varies at different stages of the pandemic, it supports previous research demonstrating the relationship between increased rates of movement, which we measure as the percentage of travel outside of local areas, and increased COVID-19 incidence.

### Network structure

To quantify how the structure of the overall network changed through time, we computed the edge betweenness centrality of connections between cells for the weighted movement networks, a measure of the relative importance of a given connection in the network (Fig 3A and 3B). Overall, the network experiences a reduction in travel following the announcement of movement restrictions in the first national lockdown, introduced on March 23, 2020 (Fig 3C). Comparing the period preceding the first national lockdown between March 10 and March 22, 2020, with the period immediately following, from March 23 to April 4, 2020, we compared which edges remained in the same betweenness quantile during the intervention period (Fig 3D). We observe a disproportionate reduction in edges which were highly central before the introduction of national restrictions, reflecting the reduction of travel along long distance connections (Fig 3E and 3F) which tend to be highly central in the network, and an increased fragmentation of the network.

While we continued to observe a weekly trend of increased between-tile movements during weekdays, the variance of weekly between-tile connections decreased during the period of national interventions (S7 Fig). Overall, the network was reduced by 10,690 edges, with only 46 new edges observed in the post-intervention period. All of these newly created edges had a distance less than 47km. The network also became more disconnected, with the number of components increasing from 23 (largest component size: 3394) to 37 (largest component size: 2579).

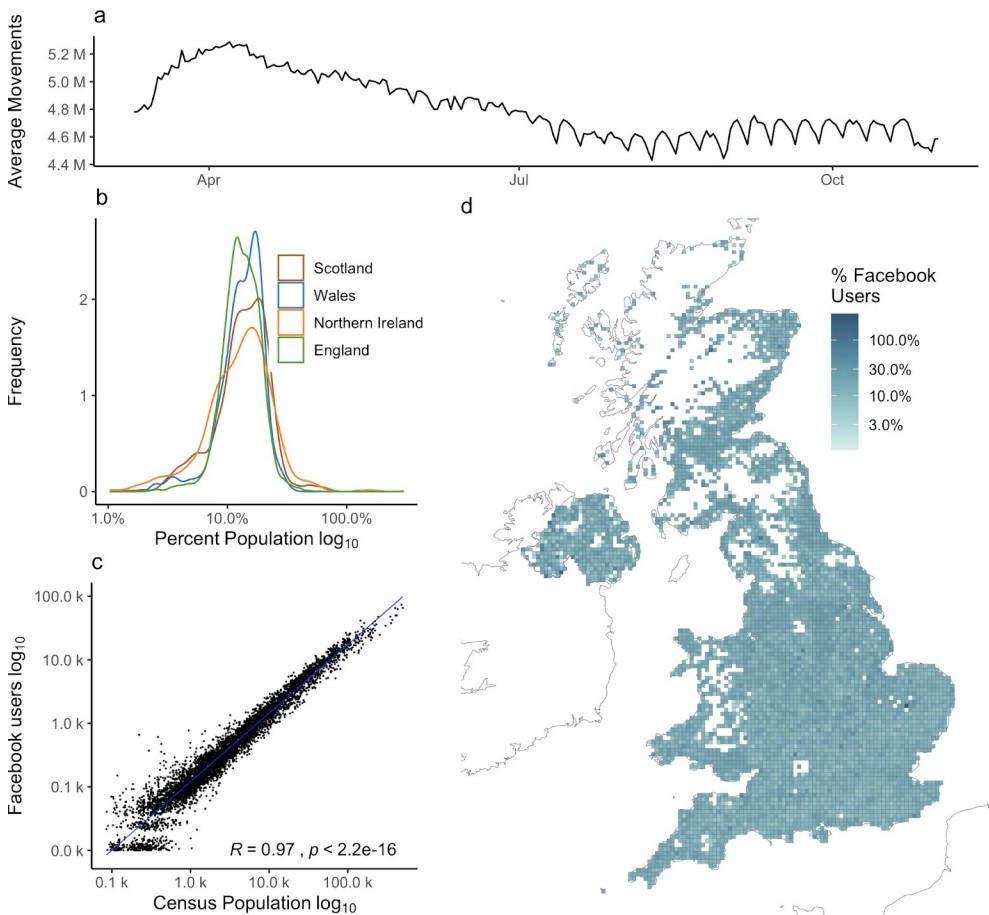

**Fig 1. Characteristics of the between-cell Facebook mobility dataset.** a) The daily average number of users included in the movement dataset. b) The probability density functions of the percentage of Facebook users in the census population for cells by country. c) The relationship between the number of Facebook users and census population for all cells. d) The spatial distribution of the percentage of Facebook users in the census population for all cells. The proportion of Facebook users to population is homogenous between regions although there are local variations in the proportion of Facebook users to population. White cells are censored due to low numbers. Note that 12 cells around the town of Swindon are missing due to a data processing error prior to data sharing. Base map data from Natural Earth [55].

We also measured the distance travelled per user in the weighted, directed network, observing a decrease in the overall distance travelled by users during the period of national interventions and a subsequent increase in this distance throughout the summer. As the overall travel in the network decreased, we observed a sharper decrease in the volume of long distance connections, with most long distance connections absent from the movement network during national restrictions (Fig 3F). This decrease reflects both the decreasing volume of long distance travel, and the increased effect of censoring during periods of lower travel volumes.

## Community detection

We identified geographically-explicit "communities of interaction" in the network of user movements using the InfoMap and Leiden algorithms (S8, S9 and S10 Figs).

We observed an increase in the number of identified communities and a corresponding decrease in their area and population size following the introduction of nationwide intervention measures on March 23rd, 2020 (Figs 4, S11 and S12). Overall, these communities were smaller

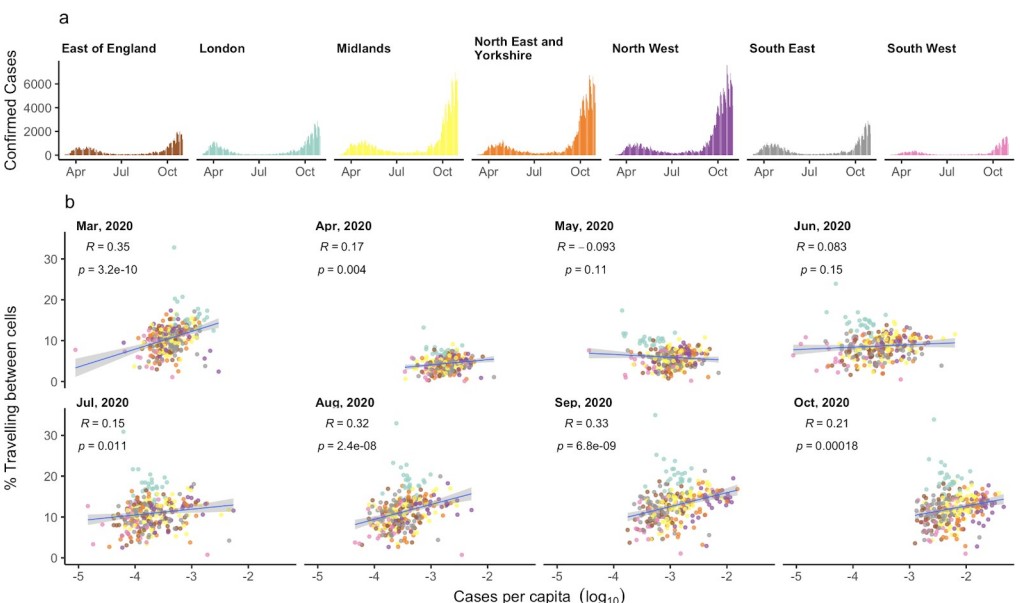

**Fig 2. The relationship between movement and COVID-19 cases per capita.** a) Daily confirmed SARS-CoV-2 cases in each NHS region. b) The relationship between the percentage of users travelling outside their cell and the total number of confirmed SARS-CoV-2 cases per capita, by month of the study period. The rate of between-cell movement is correlated with the number of COVID-19 cases and the strength of this correlation varies through time. Dots show LTLAs coloured by their NHS region.

than LTLAs and Travel to Work Areas (TTWAs) in terms of their size and population, although in cities like London and Manchester, communities intersected multiple TTWAs and LTLAs (S13 Fig). The cell-level network also became more sparse as cells were censored from the dataset due to the lower number of edges connecting cells. Restrictions were eased incrementally between May and July 2020, during which time we observed an increase in the volume of between-cell movements and an increase in the geographic area and connections between communities.

We found that the most persistent communities existed in some large population centers (S14 Fig). This reflects both the smaller influence of censoring on higher population cells as well as the continued existence of movement networks, though reduced, around population centres. Persistent communities were identified in Manchester, Newcastle, Glasgow, and Edinburgh, but not in London, which regularly split into more communities on weekends (S15 Fig). We did not find evidence that community stability is associated with population density (S16 Fig). We found strong stability of communities calculated for networks aggregated to daily, weekly, and monthly intervals, with a Normalised Mutual Information, a measure of similarity between network partitions, between 0.89 and 0.92 for all partitions (S17 Fig).

By transferring community labels between time windows, we constructed a network of communities in which each community is a node, connected to other communities in a directed network weighted by the number of users travelling between community pairs (Fig 4). In this network, the degree of all nodes decreased after the implementation of nationwide interventions and the overall reduction of between-cell edges (S18 Fig). Nonetheless, we did not observe a significant reorganisation in the hierarchy of connections between communities.

## Local lockdown extents

After the period of national interventions, the UK introduced local area interventions at differing levels of stringency. The first such intervention was implemented in Leicester on June 30,

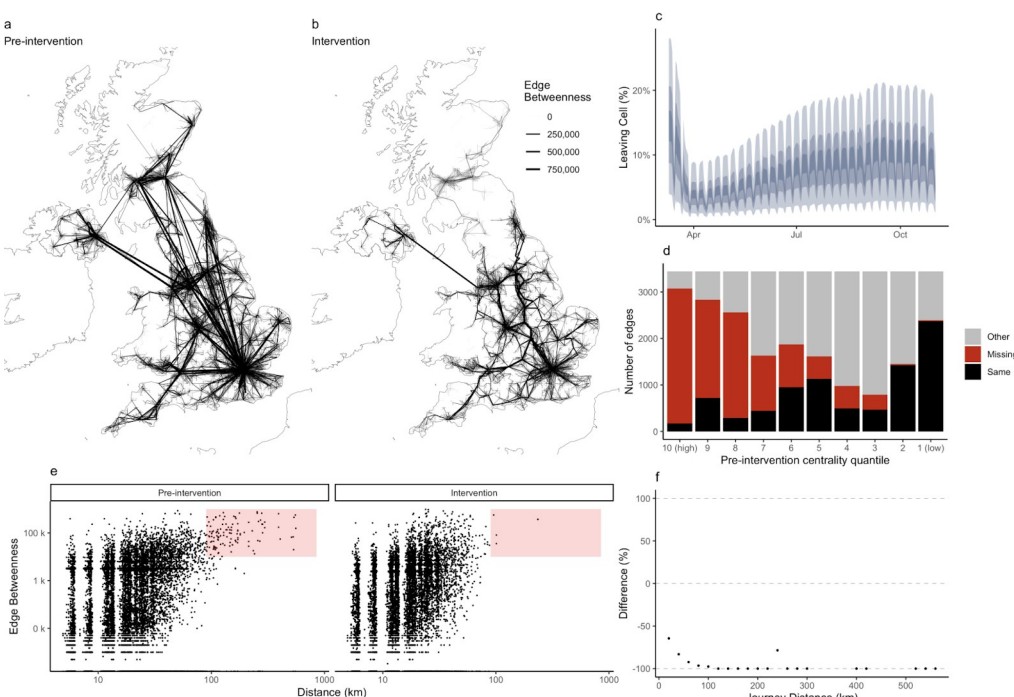

**Fig 3. Network structure through time.** All network connections in a) the period preceding the first national lockdown (March 10 to March 22, 2020) and b) following the first national lockdown (March 23 to April 4, 2020). The thickness of edges corresponds to their edge betweenness centrality. c) The percentage of users travelling out of their cell through time, shown as 90%, 75%, and 50% density intervals; and d) the edges that remain highly central after the introduction of national interventions. We compare the betweenness quantile for edges from March 10 to March 23, 2020 (preceding national interventions) and from March 23 to May 10, 2020 (period of national interventions). This shows a disproportionate elimination of network connections for edges which were highly central before the announcement of the first national lockdown. e) The distribution of edge betweenness and distance pre- and post-intervention on March 23rd. The red highlight demonstrates the loss of long-distance, high betweenness edges. f) The reduction of travel by distance during national interventions. Most long-distance edges are removed from the dataset entirely as a result of decreased travel. Note that the longest remaining connection is between Liverpool and Belfast. Base map data from Natural Earth [55].

2020 in response to a local resurgence. To understand the impact on the mobility of users, we assessed changes in the volume of travel and network topology before and after introduction.

We measured the connection between cells overlapping areas of local interventions in four areas: Leicester, Manchester, the North West, and the North East (S19, S20 and S21 Figs) to assess the impact on volume of travel and the isolation of intervention areas from the broader UK movement network. We found that, while movement indicators did decrease, particularly in Leicester, the response was smaller than during the first national lockdown (Fig 5A). The introduction of local interventions measures in Leicester was followed by a reduction in cases in the area, asomething that was not observed for the other local interventions (Figs 5B, S19, S20, and S21).

Motivated by the need to identify communities associated with epidemic resurgences and responses to reactive interventions, we compared the extent and date of local interventions with the spatial extent and temporal persistence of network communities (Fig 5C). We found that network communities remained relatively stable after the introduction of intervention measures in all areas, with some peripheral changes to movement communities in Manchester, the North West, and North East (S19, S20 and S21 Figs).

From July 2020 onwards, the geographic extent of local area interventions was in closer agreement with movement communities, particularly in Manchester (S19 Fig). Some

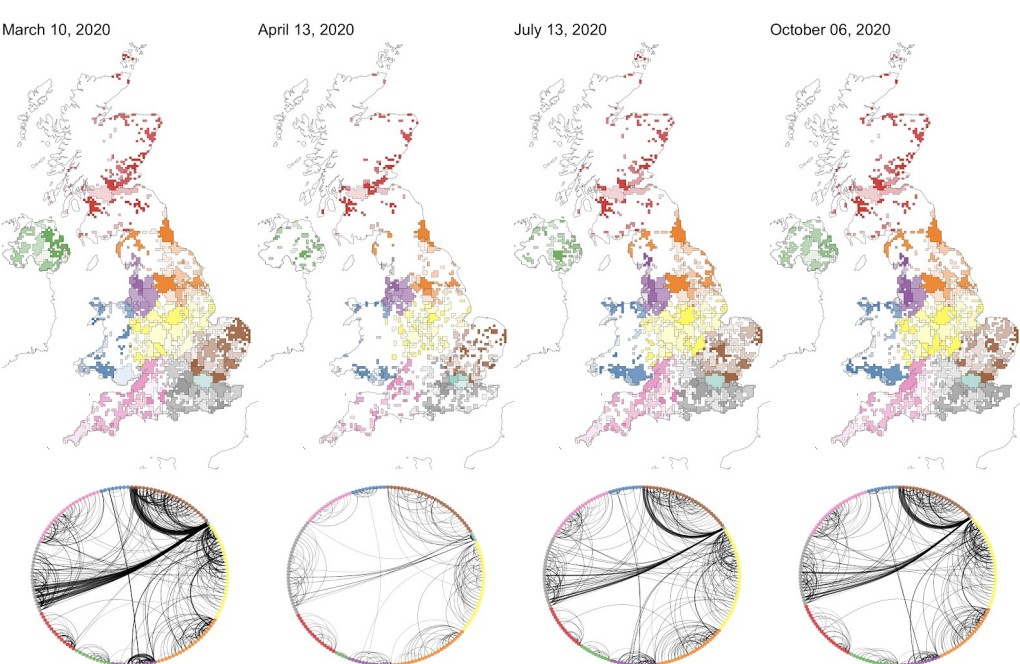

**Fig 4. Community detection using the InfoMap algorithm.** Communities detected on individual days through the time series. Areas delimit the maximum extent of all cells in detected communities and are colored based on the NHS region they intersect. Network plots show the volume of travel between communities for the network of connections between communities on each day. Missing tiles which record fewer than 10 users moving have been censored for privacy and appear white. The number of communities on each day (left to right): 189, 288, 218, 209. Base map data from Natural Earth [55].

interventions also spanned multiple movement communities, as in the North West and North East of England (S20, and S21 Figs). Early local intervention measures at limited spatial extents may not have fully encompassed the area of transmission resurgence, however UK policy changed over time, beginning to enforce collaborative local area interventions comprising

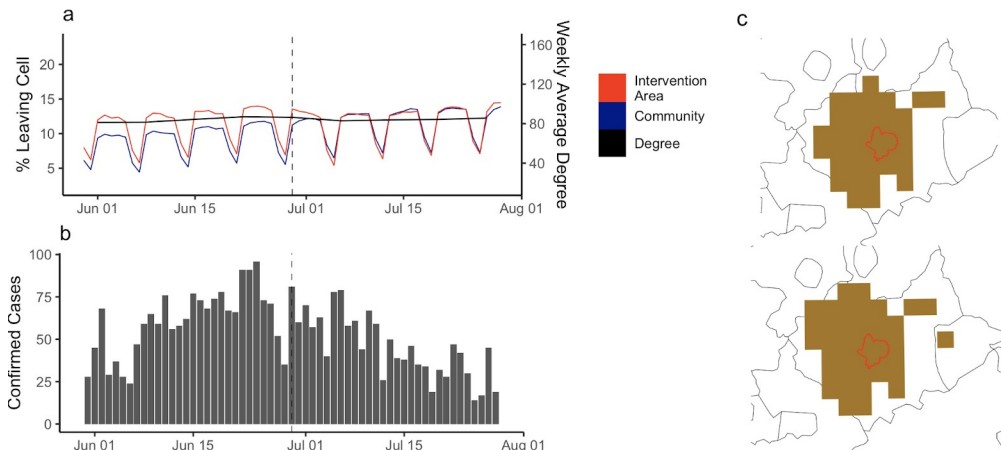

**Fig 5. Response to the introduction of local interventions.** a) The daily percentage of users travelling between cells in the intervention area and the connected community, and the weekly average degree of intervention cells. This shows the change in movement indicators through time, as well as the change in connectivity of this community to the remaining network of communities. b) Confirmed cases in the intervention area. c) Changes in the community structure before and after the introduction of local intervention. Base map data from Natural Earth [55].

multiple Local Authorities. Additionally, movement communities evolve over time, and have the potential to shift following local area interventions, requiring an understanding of real-time patterns of movement to monitor the appropriateness of a given measure.

## Discussion

This study used a large, anonymized movement dataset to quantify changes in the UK movement network and assessed how geographic communities were affected by interventions. Using movement communities, we can identify strongly connected areas that can be used to inform the spatial scale of geographically-targeted interventions to respond to resurgences of COVID-19. These communities can also be used to monitor adherence to public health measures in near-real time, as changes in community structure indicate a reorganization of the patterns of travel for specific populations. We found that overall, these communities were smaller in population and area than LTLAs and TTWAs, providing a finer-grained understanding of mobility connections between areas. We also explored the structure of the UK travel network through the pandemic, identifying variations in the central connections between population centres and changing travel patterns in response to the introduction of public health measures.

Gridded mobility datasets such as those used in this study provide granular, near real-time information about the movement patterns of a large sample of the UK population. While these datasets could usefully inform epidemic responses [18,44–46], there remain questions about the generalizability of the movement recorded in these datasets to the movements of the overall population [47,48]. The privacy preserving structure of the Facebook movement dataset means that low frequency connections are not recorded, precise locations are replaced by grid cell references, and data is provided in uniform-area grid cells which vary in population size. Ideally, multiple mobility datasets should be analysed to improve the interpretation of changes in mobility indicators.

In response to the nationwide lockdown introduced March 23rd, the UK mobility network changed drastically. The movement network became more sparse, with reductions in travel volume and distance. These changes disproportionately affected long distance and highly central edges, important connections which integrated geographically distant areas in the broader UK travel network.

We identified geographically distinct communities with strong interconnections that are relevant to policy responses focused on limiting transmission in response to geographically-limited disease resurgences. These communities delimit the boundaries of areas with strong internal movement connections and therefore provide empirical boundaries that can inform policy responses or form the basis for spatially-explicit transmission models. Community structure has been shown to influence the progression of modelled disease transmission in simulated and real-world networks [49–51]. Variations in community structure can affect or limit mixing between people or between populations, giving rise to epidemics with multiple peaks or geographically-localised resurgences [52]. Identifying communities is therefore important for understanding and projecting epidemic dynamics.

There is a demonstrated relationship between human mobility and transmission of COVID-19 relating both to the spatial introduction of disease, where connectivity is correlated with the time of first introduction [15], and rates of movement in specific locations, which are correlated with disease transmission [53]. In the movement dataset used in this study, we identified groups of network nodes which have strong movement connections to each other, meaning that they share a relatively high number of travellers with other members of their community. Similar delimiting of movement communities has been applied to define

"functional zones" in the European Union [54], where these zones have been recommended as a way to define the extent of locally targeted interventions when responding to spatially-limited resurgences of COVID-19.

In response to resurgences in a particular area, determining the geographic extent of reactive interventions should be driven by areas at risk of increased transmission, which may not intersect with administrative boundaries. The geographic areas identified here could be used to delimit the extent of areas with strong connections to a particular resurgence or new outbreak and to define the extent of interventions such as surge testing or social distancing measures. We found that while communities tended to stabilise around settlements, there was disagreement between the extent of these communities and the boundaries at which local area interventions have been introduced in the UK thus far. These communities provide valuable information in near-real time about the extent of typical patterns of travel, their temporal variations, and "catchment areas" of movement around a given area.

There are several caveats to the methods of community detection used in this study, as the extent of communities could be influenced by the level of aggregation of the Facebook mobility data, and cells were assigned to a single community each day. While we conducted a sensitivity analysis using two methods for identifying communities, there are a wide variety of community detection algorithms which emphasize different aspects of network structure. Questions also remain about the general reliability of community detection methods, which have been developed on well understood network structures, when applied to real-world networks [35]. The effect of local area interventions on travel depends on the specifics of each intervention and their stringency. Additionally, interventions occur at multiple spatial scales, and across overlapping time periods. For example, in the UK, national interventions coincide with local interventions, and each may contribute differently to changes in movement behaviour.

## Conclusion

Data-driven approaches using mobility data can help to quantify patterns of travel and inform geographically-targeted public health interventions.

## Supporting information

**S1 Fig. An overview of different geometries used in this study, intersecting London.** a) Zoom level 12 tiles, b) Zoom level 13 tiles, c) Lower Super Output Areas. Base map data from Natural Earth and Office for National Statistics [55–56].
(TIF)

**S2 Fig. The number of cells included in the mobility dataset (zoom level 12).** Cells recording fewer than 10 persons moving between cells along any connection are censored from the dataset to preserve user privacy.
(TIF)

**S3 Fig. The cell-level geographic distribution of the four census variables used in the analysis.** In each case, a white cell means the data were missing from the Facebook mobility data and so are not displayed here. In most cases this is due to censoring of low numbers, except for the small discontinuity around Swindon, mentioned in the Main Text. a) IMD rank. Each country has a different colour because the measure of IMD is different in each country. In each case, the darker shade is higher IMD. b) Population density per cell (log scale). c) Percentage of the population self-identifying as any other ethnicity than "Any white background". d) Mean age of the population resident in each cell. Base map data from Natural Earth [55].
(TIF)

**S4 Fig. A comparison of the percentage of Facebook users and census variables at the cell level.** a) IMD, b) mean age, c) percent minority ethnic, and d) population. Variables were aggregated from mid-level census geographies for each country. The mean value of each variable was assigned to intersecting tiles, weighted by small area population estimates. Correlation is shown on the panel as the Pearson correlation coefficient ($R$) and two-sided p values.
(TIF)

**S5 Fig. A comparison of movement by IMD quintile.** Percent change from baseline for movement between cells by IMD quintiles in each country. IMD data was aggregated to cell level and weighted by small area population estimates. IMD quintiles range from 1 (most deprived) to 5 (least deprived).
(TIF)

**S6 Fig. The relationship between movement and COVID-19 cases.** a) Daily confirmed SARS-CoV-2 tests in each NHS region. b) The relationship between the percentage of users travelling outside their cell and the total number of confirmed SARS-CoV-2 positive tests, by month of the study period. Dots show lower-tier local-authorities coloured by their NHS region as in panel a.
(TIF)

**S7 Fig. Weekly variance of between-cell movement.** The standard deviation of between cell movements through time. Decreased variance indicated smaller differences in daily between-cell travel measurements per week. This reflects a reduction in the weekly pattern of between-cell movements.
(TIF)

**S8 Fig. Spatial comparison of community detection algorithms.** The extent of communities detected by InfoMap (a) and Leiden (b) on March 19th. Leiden communities are largely a superset of communities detected by Infomap, indicating the detection of a different hierarchical structure, but an agreement of community boundaries between the two algorithms. Base map data from Natural Earth [55].
(TIF)

**S9 Fig. InfoMap and Leiden community detection methods.** A comparison of communities detected with the InfoMap (red) and Leiden (black) methods for a selection of dates. Base map data from Natural Earth [55].
(TIF)

**S10 Fig. Comparing community detection algorithms.** A comparison of the spatial intersection between communities detected by the Infomap and Leiden algorithms through time. a) The average % overlap of InfoMap communities with Leiden communities. b) The number of Leiden communities each InfoMap community intersects, and c) the inverse comparison of Leiden communities to InfoMap communities. For a specific date, d) the maximum % areal overlap of each InfoMap community with all Leiden communities. e) The number of Leiden communities each InfoMap community intersects, and c) the inverse comparison of Leiden communities to InfoMap communities. This shows spatial alignment between InfoMap and Leiden communities, where Leiden communities tend to be larger than those detected by Info-Map.
(TIF)

**S11 Fig. Number of InfoMap communities.** The number (red) of InfoMap communities through time. An increase in the number of communities reflects more local patterns of travel

during national interventions.
(TIF)

**S12 Fig. Area and population of InfoMap communities.** a) The average and median area of InfoMap communities through time. b) The average and median census population of communities through time. The distribution of population is more skewed than that of area, reflecting high populations in specific communities.
(TIF)

**S13 Fig. Comparison of TTWAs, LTLAs, and detected communities.** Maps of 2011 TTWAs, LTLAs, and InfoMap communities detected on a specific date. While some InfoMap communities are similar in size to TTWAs and LTLAs, they do not share the same boundaries. There are fewer communities (193) than TTWAs (228) and LTLAs (380) and these communities have a smaller average area across all time periods (389.73 km$^2$) compared to TTWAs (1069.14 km$^2$) and LTLAs (605.28 km$^2$). Base map data from Natural Earth and Office for National Statistics [55,57,58].
(TIF)

**S14 Fig. Persistence of communities.** a) the total number of community labels that each cell has had (i.e. number of communities that the cell has ever been in) during the study period. The darkest shade indicates that a cell was always in the same community. b) the number of community labels for a given cell as a proportion of the number of days that cell was present in the dataset. This was calculated as the (number of unique community labels/number of days a cell was present). c) stable communities, marked as those which had the same community label for the entire study period. Base map data from Natural Earth [55].
(TIF)

**S15 Fig. Community persistence in the dataset.** a) The most persistent communities, those that existed throughout the timeseries, as on March 19th, 2020. b) Community membership from March 19th to March 26th, 2020 in Leicestershire, and c) community membership from March 19th to April 5th, 2020 in London. For both figures, panels are ordered row-wise from top left. Base map data from Natural Earth [55].
(TIF)

**S16 Fig. The relationship between community labels and population density.** The relationship between the population density of individual cells and the number of community labels assigned during the period. We do not observe a strong association between population density and the number of community labels assigned to individual cells.
(TIF)

**S17 Fig. Community detection with different time aggregations.** Normalised Mutual Information, a measure of similarity between community partitions, measured for InfoMap communities detected in networks aggregated to daily, weekly, and monthly periods. The partition of each network is compared to its parent time aggregation (days to overlapping weeks, weeks to overlapping months).
(TIF)

**S18 Fig. The relationship between community degrees during the first national lockdown.** The total degree of movement communities assigned to individual tiles. We observe a relationship between the degree of communities before and during national interventions, indicating that highly connected communities remained highly connected during the period of national interventions. The dashed line indicates where the Pre-lockdown degree is equal to the Post-

lockdown degree. All points would fall along this line in the event of no changes to the community network after the intervention.
(TIF)

**S19 Fig. Communities connected to Manchester local area restrictions.** The daily percentage of users travelling between cells in the intervention area and the connected community, and the weekly average degree of intervention cells (a). Confirmed cases in the intervention area (b). Changes in the community structure before and after the introduction of local intervention (c). Base map data from Natural Earth [55].
(TIF)

**S20 Fig. Communities connected to North East England local area restrictions.** The daily percentage of users travelling between cells in the intervention area and the connected community, and the weekly average degree of intervention cells (a). Confirmed cases in the intervention area (b). Changes in the community structure before and after the introduction of local intervention (c). Base map data from Natural Earth [55].
(TIF)

**S21 Fig. Communities connected to North West England local area restrictions.** The daily percentage of users travelling between cells in the intervention area and the connected community, and the weekly average degree of intervention cells (a). Confirmed cases in the intervention area (b). Changes in the community structure before and after the introduction of local intervention (c). Base map data from Natural Earth [55].
(TIF)

## Acknowledgments

The following authors were part of the Centre for Mathematical Modelling of Infectious Disease COVID-19 working group. Each contributed in processing, cleaning and interpretation of data, interpreted findings, contributed to the manuscript, and approved the work for publication: Sam Abbott, Kaja Abbas, Kiesha Prem, Sebastian Funk, Jon C Emery, Georgia R Gore-Langton, Fiona Yueqian Sun, Arminder K Deol, Alicia Showering, Nicholas G. Davies, Nikos I Bosse, Samuel Clifford, Anna M Foss, Graham Medley, C Julian Villabona-Arenas, Timothy W Russell, Amy Gimma, W John Edmunds, Gwenan M Knight, Yung-Wai Desmond Chan, Yalda Jafari, Quentin J Leclerc, Rein M G J Houben, Akira Endo, Sophie R Meakin, Petra Klepac, Joel Hellewell, Naomi R Waterlow, Kevin van Zandvoort, Christopher I Jarvis, Rachel Lowe, Matthew Quaife, Charlie Diamond, Megan Auzenbergs, Simon R Procter, Rosanna C Barnard, Oliver Brady, Katherine E. Atkins, Katharine Sherratt, Thibaut Jombart, Stéphane Hué, Kathleen O'Reilly, Jack Williams, David Simons, Stefan Flasche, Mark Jit, James D Munday, Billy J Quilty, Frank G Sandmann, Damien C Tully, James W Rudge, Alicia Rosello.

## Author Contributions

**Conceptualization:** Hamish Gibbs, Yang Liu, James Cheshire, Leon Danon, Chris Grundy, Adam J. Kucharski, Rosalind M. Eggo.

**Data curation:** Hamish Gibbs, Emily Nightingale, Rosalind M. Eggo.

**Formal analysis:** Hamish Gibbs, Emily Nightingale, James Cheshire, Rosalind M. Eggo.

**Funding acquisition:** Chris Grundy, Adam J. Kucharski, Rosalind M. Eggo.

**Investigation:** Hamish Gibbs, Emily Nightingale, Yang Liu, James Cheshire, Leon Danon, Liam Smeeth, Carl A. B. Pearson, Chris Grundy.

**Methodology:** Hamish Gibbs, Emily Nightingale, Yang Liu, James Cheshire, Leon Danon, Liam Smeeth, Carl A. B. Pearson, Chris Grundy, Adam J. Kucharski, Rosalind M. Eggo.

**Project administration:** Hamish Gibbs, Adam J. Kucharski, Rosalind M. Eggo.

**Resources:** Chris Grundy, Adam J. Kucharski, Rosalind M. Eggo.

**Software:** Hamish Gibbs, Emily Nightingale, Yang Liu, James Cheshire.

**Supervision:** James Cheshire, Leon Danon, Liam Smeeth, Chris Grundy, Adam J. Kucharski, Rosalind M. Eggo.

**Validation:** Hamish Gibbs, Emily Nightingale, Yang Liu, James Cheshire, Leon Danon, Liam Smeeth, Carl A. B. Pearson, Chris Grundy, Adam J. Kucharski, Rosalind M. Eggo.

**Visualization:** Hamish Gibbs, James Cheshire.

**Writing – original draft:** Hamish Gibbs, Adam J. Kucharski, Rosalind M. Eggo.

**Writing – review & editing:** Hamish Gibbs, Emily Nightingale, Yang Liu, James Cheshire, Leon Danon, Liam Smeeth, Carl A. B. Pearson, Chris Grundy, Adam J. Kucharski, Rosalind M. Eggo.

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
