## [Decision Letter · Decision Letter 0]

19 Apr 2021

Dear Mr. Gibbs,

Thank you very much for submitting your manuscript "Detecting behavioural changes in human movement to inform the spatial scale of interventions against COVID-19." for consideration at PLOS Computational Biology.

As with all papers reviewed by the journal, your manuscript was reviewed by members of the editorial board and by several independent reviewers. In light of the reviews (below this email), we would like to invite the resubmission of a significantly-revised version that takes into account the reviewers' comments.

We cannot make any decision about publication until we have seen the revised manuscript and your response to the reviewers' comments. Your revised manuscript is also likely to be sent to reviewers for further evaluation.

Sincerely,

Alex Perkins

Associate Editor

PLOS Computational Biology

Virginia Pitzer

Deputy Editor-in-Chief

PLOS Computational Biology

Reviewer's Responses to Questions

**Comments to the Authors:**

Reviewer #1: In this work, the authors use human mobility data from Facebook users in the UK to analyze the associated spatial network of movements before, during and after the national lockdown and in the period of local interventions. They applied community detection techniques and measured the evolution of the detected communities over time. These analyses revealed that the mobility network became sparser, the number of communities was decreased, and long-distance journeys were less frequent during the lockdown. Local interventions did not change the community structure and only decreased travel outside of local areas weakly.

The authors address a highly relevant topic with potentially big implication for public health and future implementations of non-pharmaceutical interventions against the spread of infections diseases. The authors make sure that their data set is not biased by comparing the local fraction of Facebook users between regions and checking for correlations with census variables.

However, I have reservations regarding the interpretation, implications, and novelties of this study. The paper lacks a clear story and does not live up to its title, because I cannot identify the “spatial scale of interventions against COVID-19”.

Major observations

1. The correlation between movement and new COVID-19 cases was reported in many previous studies, eg in

Badr, Hamada S., et al. "Association between mobility patterns and COVID-19 transmission in the USA: a mathematical modelling study." The Lancet Infectious Diseases 20.11 (2020): 1247-1254.

This should be at least mentioned, if not discussed.

2. To quantify changes in the network structure the authors calculate the edge betweenness of connections. It is unclear whether the authors assigned weights to the links when computing this quantity. If not, it seems to me that only the removal of links due to the censoring of links with less than 10 travelers changes the observable. Also, the motivation for this observable is not clear to me. What do we learn about behavioural changes from it, additionally to just quantifying the length and frequency of trips?

3. Next, in the central part of the paper, the authors identify communities in the network and, subsequently, a corresponding network of communities, with links weighted by travelers. Here, unexpectedly, the study finds that the average node degree decreased during lockdown. However, any real analysis of this community network relevant to disease spread is missing. The same is true for the obtained community distribution. Do they match political organizational units or commuting zones? Do they have typical size?

The network could also serve as the basis for an agent-based or multiscale models of disease spread, which is not discussed.

4. Lastly, the authors analyze the effect of local interventions. Here the main result is that the effect of local interventions was weaker than a national lockdown. Also, the extent of communities did not always match the boundaries of local interventions in the UK. It is unclear if the authors suggest using their identified communities as boundaries for local interventions. I am missing a clear evidence that the proposed strategy of identifying communities using mobility data can improve interventions against COVID-19.

5. Captions of all figures are too short and do not guide the reader to quickly grasp the message of the figure. Two examples:

a. Fig1: The caption only repeats the axes labels but does not contain the message: that the number of travelers is positively correlated with new covid cases!

b. Fig2: Again, it seems to me that the message of the figure is that the percentage of Facebook users is quite homogeneous between regions and can therefore be used to say something about the full population.

Minor observations

1. Caption of fig 1. Speaks of daily reported SARS-CoV-2 tests in a) but the figure shows confirmed cases.

2. Caption of fig 2. Speaks of daily average number of users in a) but the axis label speaks about average movements. Also, areas with more than 75% of the population registered as users were removed. Why?

3. Fig 3d is absolutely unclear to me. The axis label says “number of edges”. The caption speaks about the likelihood of a journey to remain highly central in the lockdown. How do you read this?

4. Fig 4. Is missing labels a),b)

5. Fig 5 is also quite difficult to understand. Clarify message in the caption. Maybe zoom in into the relevant date range for the local interventions? Maybe show the number of travellers in the intervention area relative to the corresponding community, otherwise it is quite hard to see a decrease here.

6. Fig S6. What is IMD?

7. Why is Fig S7 not in the main manuscript? It seems to be quite central as it tells us about the spatial scale of movement in the lockdown (max 50-100 km).

8. Same with Fig S8. It makes it clearer, which journeys are removed in the lockdown. Also, add small ticks to this plot to make the log-log axes clearer and easier to read.

9. Fig S11 is unclear to me. What should the reader see?

10. In Fig S16, what is the dashed line? It does not seem to be a fit.

11. The message of Figs S17-19 is also unclear. They could also benefit from zooming in the time series into the relevant time periods.

Reviewer #2: The paper investigates the emerging reorganisation of mobility network during the first wave of COVID-19 in UK by means of a large-scale fine-grained mobility dataset.

The paper is well written, easy to follow and understand, even if I think that might further improve with a slightly different structure (details below). The Introduction and Methods sections provide a suitable context for the work, but some relevant references are missing (details below). The figures are high quality and appropriate for illustrating the ideas and work presented. Overall, the analysis is sound, the paper is self-contained and presents a suitable selection of results to support the main findings.

While I think the paper is interesting and has merit, it mainly reports a descriptive data analysis, therefore it will greatly improve if the authors will widen the discussion and implications of their findings in the context of public health interventions.

Major comments

- Relationship between movement and COVID-19 cases: corroborated by the correlations shown in Figure 1, the authors suggest that there is an “intuitive relationship between increased rates of travel outside of local areas and increased COVID-19 prevalence”. This statement needs better framing and a deeper discussion. First, the volume of travels outside a local area would be relevant for the local spreading only under the hypothesis that such travels are mainly due to commuting patterns that come back to the local area of origin. While this might be true for the data used in this paper, some discussion about this point is needed. Secondly, it is well known that -in the context of infectious diseases spreading-, travel fluxes are associated with the arrival time of the disease in not-yet-affected areas [Brockmann2013, Balcan2009]. However, after the initial seeding the local outbreak unfolds with an exponential growth and thus further importation of cases should not play a major role in shaping the epidemic curve. Therefore, the author should elaborate on this concept to let the reader grasp their intuition, and why such result is interesting. Finally, I think that a normalized quantity for the number of cases (as reported in Supplemental Figure 3) would be a more suitable observable, as the raw count of cases is clearly influenced by the number of people living in certain area.

- Characteristics of the between-cell Facebook mobility dataset: the authors considered “the percentage of Facebook users per cell […] homogeneous across the study area (Figure 2d)”. However, Figure 2B shows distributions with an average value ~10% and long tails up to ~70%, that seems quite heterogeneous. The authors should clarify or better frame this point. Moreover, the correlation reported in Figure 2C would definitely benefits from a log-log transformation, to limit the impact of few outlying data points in skewing the estimate of the correlation coefficient.

- Network Structure: the authors compared the betweenness quantile for links of the mobility networks extracted before the national interventions (from March 10 to March 23) and during the national interventions (from March 23 to May 10). My first concern is that the two networks are not comparable as they are built aggregating 13 days and 47 days respectively. Therefore, I would expect some major differences even in the absence of exogenous interventions. In principle, a longer aggregation window might result in a denser network (light links from rare travels are reinforced with longer observation time). The authors might consider to perform the analysis on mobility networks extracted on comparable time windows. Moreover, the community detection analysis seems to be performed on daily mobility networks: relying on the same temporal aggregation would help the reader compare the findings of the different network analysis.

- Since the mobility network is largely destroyed by the national interventions, besides the information on betweenness quintiles, some coarser information about mobility network resilience could be helpful to better understand how the system evolved (e.g. how many links of the pre-intervention network are preserved in the post-intervention period? How many new links were created, if any?).

- Some communities detected on mobility networks of individual days are quite stable across time. A natural question is to verify if such pattern persists also when mobility data are aggregated at different temporal scales (e.g. weekly, monthly,…).

- The authors show how the community structure in four areas (Leicester, Manchester, the North West, and the North East) seems to be stable before and after some local interventions, with the only exceptions of some grid cells. There has been some recent work in assessing the association between socio-economic status and change of mobility induced by COVID-19. I am aware that it is rather uncouth of me to suggest referencing my own work, but our paper www.medrxiv.org/content/10.1101/2020.11.16.20232413v1 seems particularly appropriate here as it explore such phenomenon at the microscale of city neighborhoods. I would not expect the authors to implement an extensive analysis in the current paper, but it would be great if they could add some insights about deprivation levels of the cells that change community membership and in general it should certainly be considered as further work. The authors could also consider to widen their discussion to include some missing reference in this direction, such as [Jay2020, Pullano2020, Gauvin2020]. Overall, the authors might consider elaborating a bit on this point, as the only analysis performed is in Supplemental Figure 6 and poorly commented in the main paper. I believe that this aspect could be important to strengthen the paper, as the implication of such analysis might suggest that deprived areas respond differently under top-down intervention, thus potentially limiting their effectiveness on more deprived and vulnerable population.

- The authors might be interested in reading (and probably citing) some references closely related to this work [Iacus2020, Bonato2020]. In those papers, the authors investigated the time evolution of data-driven Mobility Functional Areas (geographic zones with high degree of intra-mobility exchanges) before and after national interventions.

Minor comments

- For the sake of readability, I suggest to move Figure 2 and the considerations about the representativeness of the dataset before Figure 1, where such data are used.

- Figure 3C: an explanation about the meaning of the different colors of the shaded areas is missing.

- In the first paragraph of page 8, the authors refer to Supplemental Figure 9 as a comparison of Infomap vs. Leiden communities, but they should probably refer to Supplemental Figures 10 and 11.

Brockmann, D. and Helbing, D., 2013. The hidden geometry of complex, network-driven contagion phenomena. science, 342(6164), pp.1337-1342.

Balcan, D., Colizza, V., Gonçalves, B., Hu, H., Ramasco, J.J. and Vespignani, A., 2009. Multiscale mobility networks and the spatial spreading of infectious diseases. Proceedings of the National Academy of Sciences, 106(51), pp.21484-21489.

Jay, J., Bor, J., Nsoesie, E.O., Lipson, S.K., Jones, D.K., Galea, S. and Raifman, J., 2020. Neighbourhood income and physical distancing during the COVID-19 pandemic in the United States. Nature human behaviour, 4(12), pp.1294-1302.

Pullano, G., Valdano, E., Scarpa, N., Rubrichi, S. and Colizza, V., 2020. Evaluating the effect of demographic factors, socioeconomic factors, and risk aversion on mobility during the COVID-19 epidemic in France under lockdown: a population-based study. The Lancet Digital Health, 2(12), pp.e638-e649.

Gauvin, L., Bajardi, P., Pepe, E., Lake, B., Privitera, F. and Tizzoni, M., 2020. Socioeconomic determinants of mobility responses during the first wave of COVID-19 in Italy: from provinces to neighbourhoods. medRxiv.

Iacus, S., Santamaria, C., Sermi, F., Spyratos, S., Tarchi, D. and Vespe, M., Mapping Mobility Functional Areas (MFA) using Mobile Positioning Data to Inform COVID-19 Policies, EUR 30291 EN, Publications Office of the European Union, Luxembourg, 2020, ISBN 978-92-76-20429-9, doi:10.2760/076318, JRC121299.

Bonato, P., Cintia, P., Fabbri, F., Fadda, D., Giannotti, F., Lopalco, P.L., Mazzilli, S., Nanni, M., Pappalardo, L., Pedreschi, D. and Penone, F., 2020. Mobile phone data analytics against the covid-19 epidemics in italy: flow diversity and local job markets during the national lockdown. arXiv preprint arXiv:2004.11278.

Reviewer #3: This was a great read. It was very interesting to see fine-scaled movement data quantified and associated in this way. I think the content stands on its own, but could be made more compelling by a stronger connection between movement data and virus reproduction rates.

General

I find myself wondering about the relationship between cell-to-cell travel and transmissible contact rates. Is it possible to characterize the types or proportion of trips that are captured by these movement data (for the average/median individual)? Commutes to work or school, trips to the grocery, etc. For instance there might be statistics on commuting that would allow you to estimate the proportion of commutes that leave a level-13 cell.

Introduction

Perhaps remove “, and network service providers” from the last sentence of first paragraph. Repeated ‘and’ and ‘network providers’.

ONS data citation?

Materials and Methods

Facebook Data: From the text it is difficult to determine if the movement data is (1) simply a geolocation taken every 8 hours, (2) change of map cell at any time during each 8 hour period, or (3) something else. If it is (2), can an individual have multiple journeys in a single 8 hour period?

Facebook Data: is there any latency or is the data for a given day released the following day?

S2 Fig caption: are these level 12 or 13 cells? For what administrative boundary? I assume for all of the U.K., but please clarify.

Demographic information: The first sentence is accurate, but a lot to unpack. I had to read through this section, look at fig S5 and then go back and reread the first sentence to fully understand it. I would suggest leading with “to identify relationships between the percentage of Facebook users and demographic factors, we…” or somesuch. Then describe the details. This is only a readability suggestion, I respect the authors’ preference.

Community Detection: What is the cost/objective function? Is there a penalty in the algorithms for adding additional communities?

S10 Fig: If it is relatively easy, it would be interesting to draw the outline of Leiden communities over the InfoMap communities.

COVID-19 Data: I do not believe LTLA is previously defined

Results

S13 Fig b and c: are these ordered row-wise or column-wise?

Discussion/Conclusion

In addition to using these methods for geographic definition of interventions and an indicator of higher transmission, I would consider discussing how these metrics could be used as real-time feedback concerning the effectiveness of intervention mandates.

A stronger connection between transmission and movement data would be compelling. It would be interesting to compare the movement data to effective reproduction number estimates and check for temporal correlation. This would remove transmission lags and ‘ramp-up’ from the comparison to cases and potentially make for a more direct connection to transmission dynamics. Is this something you considered?

**Have the authors made all data and (if applicable) computational code underlying the findings in their manuscript fully available?**

Reviewer #1: **No: **Data used in this study are available from the Facebook Data for Good Partner

Program by application. The computational code is not available.

Reviewer #2: **No: **Data used in this study are available from the Facebook Data for Good Partner Program by application.

Reviewer #3: Yes

PLOS authors have the option to publish the peer review history of their article (what does this mean?). If published, this will include your full peer review and any attached files.

Reviewer #1: No

Reviewer #2: **Yes: **Paolo Bajardi

Reviewer #3: No
---

## [Editor Report · Decision Letter 1]

5 Jun 2021

Dear Mr. Gibbs,

We are pleased to inform you that your manuscript 'Detecting behavioural changes in human movement to inform the spatial scale of interventions against COVID-19.' has been provisionally accepted for publication in PLOS Computational Biology.

Best regards,

Alex Perkins

Associate Editor

PLOS Computational Biology

Virginia Pitzer

Deputy Editor-in-Chief

PLOS Computational Biology

---

## [Editor Report · Acceptance letter]

5 Jul 2021

PCOMPBIOL-D-21-00363R1 

Detecting behavioural changes in human movement to inform the spatial scale of interventions against COVID-19.

Dear Dr Gibbs,

I am pleased to inform you that your manuscript has been formally accepted for publication in PLOS Computational Biology. Your manuscript is now with our production department and you will be notified of the publication date in due course.

With kind regards,

Zsofi Zombor
